# Association of HDL Subfraction Profile with the Progression of Insulin Resistance

**DOI:** 10.3390/ijms241713563

**Published:** 2023-09-01

**Authors:** Peter Piko, Tibor Jenei, Zsigmond Kosa, Janos Sandor, Nora Kovacs, Ildiko Seres, Gyorgy Paragh, Roza Adany

**Affiliations:** 1Department of Public Health and Epidemiology, Faculty of Medicine, University of Debrecen, 4032 Debrecen, Hungary; piko.peter@med.unideb.hu (P.P.); jenei.tibor@med.unideb.hu (T.J.); sandor.janos@med.unideb.hu (J.S.); kovacs.nora@med.unideb.hu (N.K.); 2National Laboratory for Health Security, Center for Epidemiology and Surveillance, Semmelweis University, 1089 Budapest, Hungary; 3Department of Health Methodology and Public Health, Faculty of Health, University of Debrecen, 4400 Nyíregyháza, Hungary; kosa.zsigmond@foh.unideb.hu; 4ELKH-DE Public Health Research Group, Department of Public Health and Epidemiology, Faculty of Medicine, University of Debrecen, 4028 Debrecen, Hungary; 5Institute of Internal Medicine, Faculty of Medicine, University of Debrecen, 4032 Debrecen, Hungary; seres@belklinika.com (I.S.); paragh@internal.med.unideb.hu (G.P.); 6Department of Public Health, Semmelweis University, 1089 Budapest, Hungary

**Keywords:** insulin resistance, HDL subfraction profile, HDL-2, large HDL, cut-off points, HOMA-IR, diabetes, high-density lipoprotein cholesterol

## Abstract

Type 2 diabetes mellitus (T2DM) is a major global public health problem, as it is associated with increased morbidity, mortality, and healthcare costs. Insulin resistance (IR) is a condition characterized by disturbances in carbohydrate and lipid metabolism that precedes T2DM. The aim of the present study was to investigate the association between HDL and its subfraction profile and the progression of IR, as assessed by the Homeostatic Model Assessment for IR (HOMA-IR) index, and to define cut-off values to identify an increased risk of IR. Individuals with a HOMA-IR greater than 3.63 were considered to have IR. The HDL subfractions were separated using the Lipoprint system, which identifies ten subfractions (HDL-1-10) in three subclasses as large (HDL-L), intermediate (HDL-I) and small (HDL-S). Analyses were performed on samples from 240 individuals without IR and 137 with IR from the Hungarian general and Roma populations. The HDL-1 to -6 subfractions and the HDL-L and -I classes showed a significant negative association with the progression and existence of IR. Among them, HDL-2 (B = −40.37, *p* = 2.08 × 10^−11^) and HDL-L (B = −14.85, *p* = 9.52 × 10^−10^) showed the strongest correlation. The optimal threshold was found to be 0.264 mmol/L for HDL-L and 0.102 mmol/L and above for HDL-2. Individuals with HDL-L levels below the reference value had a 5.1-fold higher risk of IR (*p* = 2.2 × 10^−7^), while those with HDL-2 levels had a 4.2-fold higher risk (*p* = 3.0 × 10^−6^). This study demonstrates that the HDL subfraction profile (especially the decrease in HDL-2 and -L) may be a useful marker for the early detection and intervention of atherogenic dyslipidemia in subjects with impaired glucose and insulin metabolism.

## 1. Introduction

According to the latest estimates of the International Diabetes Federation (IDF), the global prevalence of diabetes (10.5% in 2021) will reach 20% by 2045, which means that 783 million adults will be living with diabetes [1]. Diabetes is a chronic disease caused by a relative or absolute deficiency of insulin, reduced insulin sensitivity of target cells, and disturbances in glycolipid and protein metabolism [2]. More than 90% of people with diabetes have type 2 diabetes mellitus (T2DM) and adults with it have a significantly higher risk of cardiovascular disease (CVD), among other diseases (cancer, blindness, kidney failure), and premature death [3]. The situation is further exacerbated by the fact that 44.7% of adult cases are undiagnosed, among whom the risk of related complications is even higher [1].

The condition that precedes T2DM is insulin resistance (IR), which is identified as an impaired biological response to insulin stimulation of target tissues, primarily the liver, muscle and adipose tissue [4]. IR impairs glucose distribution, resulting in a compensatory increase in beta-cell insulin production and hyperinsulinemia. The metabolic consequences of IR can result in hyperglycemia, hypertension, dyslipidemia, visceral adiposity, hyperuricemia, elevated inflammatory markers, endothelial dysfunction and a prothrombotic state [5].

Carbohydrate and lipid metabolisms are closely interrelated with the three main components of IR-associated dyslipidemia, namely increased triacylglycerol (TAG) levels, decreased high-density lipoprotein cholesterol (HDL-C) levels and altered low-density lipoprotein (LDL) cholesterol composition [6]. Hyperinsulinemia and central obesity, which typically accompany IR, are thought to lead to the increased hepatic secretion of large TAG-rich very-low-density lipoprotein (VLDL) particles and the impaired clearance of VLDL, resulting in elevated plasma TAG levels. These VLDL particles can undergo intravascular processing by lipases and transfer proteins, giving rise to smaller and denser LDL particles. These LDL particles have a higher affinity for arterial wall proteoglycans and are more susceptible to oxidative modification, thus increasing the risk of atherosclerosis. The presence of IR is associated with an increased risk of CVD, partly due to alterations in the lipid profile [7]. HDL is involved in these processes through its function in cholesterol transport and may therefore be a common indicator of changes in the whole lipid profile. The role of HDL function in the pathogenesis of IR (and hence T2DM) has become a topic of research [8], but the underlying mechanisms are still not fully understood.

HDL is a class of lipoprotein that carries cholesterol from the tissues to the liver for excretion or recycling. It is often called the “good” cholesterol because it protects against CVDs by removing excess cholesterol from the arteries and preventing inflammation and oxidation [9]. HDL is a heterogeneous group of lipoproteins that can be divided into different subfractions based on their size, density and composition, which have different effects on CVD risk [10,11]. The number and nomenclature of the subfractions depend on the separation method used [12]. The Lipoprint HDL system (used in the present study) can separate ten subfractions into three subclasses, namely large-HDL (HDL-L: HDL-1 to -3), intermediate-HDL (HDL-I: HDL-4 to -7) and small-HDL (HDL-S: HDL-8 to -10). HDL-L (Lipoprint) corresponds to HDL_2_ (HDL_2a_ and HDL_2b_) as defined by other methods, whereas HDL-I and HDL-S together correspond to HDL_3_ (HDL_3a_, HDL_3b_ and HDL_3c_) [13,14]. For more details, see Appendix A, which is based on the article by Rosenson et al. [14], with the addition of fractions identified by the Lipoprint HDL system.

As the first step in HDL synthesis, lipid-poor apolipoprotein AI (ApoA1—produced by the small intestine) forms the discoidal HDL_1_ (nascent pre-beta) with plasma membrane phospholipids and unesterified cholesterol [15]. From peripheral cells (macrophages, liver cells and intestinal epithelial cells), ApoA1 takes up cholesterol from the HDL particle via the ATP binding cassette subfamily A member 1 (ABCA1) transporter and converts it into (pre-beta) HDL_2_ with a higher cholesterol content. Under the action of the lecithin–cholesterol acyltransferase (LCAT) enzyme, which is mainly activated by ApoA1, its cholesterol content is converted into cholesteryl ester (CE) and gradually transforms into spherical HDL_3_, to which additional apolipoproteins and other enzyme proteins are bound [16]. Under the action of LCAT, it is converted into HDL_2a_, which is larger and has higher CE content. The HDL_2a_ particle exchanges its CE content with other lipoproteins, such as VLDL, for triglycerides via CE transfer protein (CETP) [17]. Other contributors to the further maturation of HDL are phospholipid transfer protein (PLTP), which transfers phospholipids from other lipoproteins to HDL, and hepatic lipase (HL), which is responsible for the hydrolysis of TAGs and phospholipids. All these processes lead to the formation of HDL_2b_, which is taken up by the liver by binding to the Scavenger receptor class B type 1 (SR-BI) receptor expressed on the liver surface [18]. Cholesterol is recycled, partly broken down into bile acids and excreted with the bile and partly excreted in the feces, while the other part is reabsorbed via Niemann-Pick C1-Like 1 (NPC1L1) receptors on the surfaces of intestinal epithelial cells [19]. Some of the lipid-free or lipid-poor apoA1 not taken up by the liver is taken up and filtered in the kidney via cubulin and megalin receptors in the proximal tubules [20].

Generally, larger and less dense HDL particles are more anti-inflammatory and anti-atherogenic than smaller and denser ones. HDL-L has a higher capacity to remove cholesterol from macrophages and inhibit LDL oxidation than HDL-I or HDL-S [11]. However, some studies have also suggested that smaller HDL particles may have beneficial effects on endothelial function and nitric oxide production [21]. The profile of HDL subfractions may vary depending on environmental, lifestyle [22] and genetic factors [23] and metabolic disorders [24]. For instance, obesity, diabetes and metabolic syndrome are associated with lower levels of HDL-L and higher levels of HDL-S [24]. IR has profound effects on lipid and lipoprotein metabolism, which possibly contribute to the development of T2DM and its complications. Understanding these changes may help to identify novel targets for the prevention and treatment of diabetic dyslipidemia.

The association of the TAG/HDL-C ratio [25,26,27] and the HDL subfraction profile [28,29,30] with the risk of IR has been described in several publications. However, there is a lack of studies that have established cut-off values for HDL-C and its subclasses/fractions that could predict an increased risk of IR (and thus T2DM) in future routine testing.

The present study aimed to investigate the association between the progression of IR, assessed by the Homeostasis Model Assessment of Insulin Resistance (HOMA-IR), and the HDL subfraction profile in 377 samples (240 control and 137 insulin-resistant subjects). A further aim was to identify the HDL subfraction(s) and subclass(es) most important for the development of IR and to define the cut-off values for those that indicate an increased risk of IR to support diagnostic routines.

## 2. Results

### 2.1. Characteristics of Study Groups

There were no significant differences between the IR and control groups in age, proportion of sex, Roma ethnicity and current smokers. There was a significant difference in body mass index (BMI), systolic blood pressure, fasting blood glucose, insulin, LDL-C, hemoglobin A1c (HbA1c), TAG and the proportion of patients with antihypertensive and antidiabetic medication. See Table 1 for more details.

### 2.2. Comparison of HDL Subfractionation Profile between the IR and Control Groups

The IR group had significantly lower native HDL-C levels compared to the control one (HDL-C_IR_: 1.02 mmol/L, 95%CI: 0.98–1.07 vs. HDL-C_control_: 1.25 mmol/L, 95%CI: 1.21–1.30; *p* < 0.001). HDL-L (HDL-L_IR_: 0.22 mmol/L 95%CI: 0.20–0.23 vs. HDL-L_control_: 0.34 mmol/L, 95%CI: 0.31–0.36; *p* < 0.001) and HDL-I (HDL-I_IR_: 0.53 mmol/L, 95%CI: 0.51–0.55 vs. HDL-I_control_: 0.64 mmol/L, 95%CI: 0.61–0.66; *p* < 0.001) subclasses showed significantly lower mean concentrations in the IR group than in the control one.

For the HDL subfraction profile, the levels of HDL-1 to -6 subfractions were significantly (*p* < 0.001) lower in the IR group compared to the control one after Bonferroni correction. See Figure 1 for more details.

### 2.3. Association of HDL and Its Subfractions with Fasting Insulin, Fasting Glucose Levels and HbA1c 

Native HDL-C levels showed a significant negative correlation with fasting insulin levels (unstandardized coefficient (B) = −16.55, standard error (SE): 2.79, *p* < 0.001), but not with fasting blood glucose (B = 0.28, SE: 0.23, *p* = 0.224) or HbA1c (B = −0.22, SE: 0.10, *p* = 0.028). Among the HDL subfractions, HDL-1 to -8 showed a significant negative correlation with fasting insulin levels, while HDL-7 and -8 showed a significantly positive correlation with fasting blood glucose levels. For the HDL subclasses, HDL-L and -I showed a significant negative correlation with fasting insulin levels. The HDL-7 subfraction showed an inverse significant correlation with HbA1c levels. See Table 2 for more details.

### 2.4. Association of HDL and Its Subfractions with HOMA-IR Levels and Insulin Resistance

Native HDL-C levels were significantly associated with both HOMA-IR as a continuous outcome variable (B = 5.75, SE: 1.01, *p* < 0.001) and with a reduced risk of IR (B = −1.91, SE: 0.45, *p* < 0.001). Among the HDL subfractions, HDL-1 to -6 showed a significant association with reduced HOMA-IR and HDL-2 to -6 with a reduced risk of IR. HDL-L and -I subclasses showed a significantly negative association with both HOMA-IR values and the risk of IR. See Table 3 for more details.

### 2.5. Determination of Optimal Cut-Off Points and Their Association with the Risk of Insulin Resistance

Native HDL-C levels showed the optimal sensitivity and specificity values at a cut-off point of 1.045 mmol/L (sens.: 0.696, spec.: 0.628; Youden index: 0.324). Based on Youden’s statistic, among the HDL subfractions, HDL-2 showed the optimal IR marker at a cut-off point of 0.102 mmol/L (sens.: 0.579, spec.: 0.796; Youden index: 0.375), whereas, among the HDL subclasses, HDL-L showed the optimal IR marker at a cut-off point of 0.264 mmol/L (spec.: 0.604, sens.: 0.804; Youden index: 0.407). See Table 4 for more details.

Native HDL-C levels below the defined optimal threshold (<1.045 mmol/L) significantly increased the risk of IR (odds ratio (OR) = 2.94, 95%CI: 1.71–5.06, *p* < 0.001). For the defined optimal cut-off points, HDL-1 to -6 and HDL-L and -I subclasses showed a significant association with an increased risk of the existence of IR. The strongest association (*p* = 1.00 × 10^−6^) with an increased risk of IR was shown for HDL-L, with individuals with less than the optimal value of 0.264 mmol/L having a 4.73-times higher risk of IR (95%CI: 2.53–8.86) than those with a higher value. See Table 5 for more details.

## 3. Discussion

Our aim in the present study was to investigate the association between the development and progression of IR (assessed by HOMA-IR) and changes in the HDL subfraction profile. Furthermore, we sought to determine the optimal cut-off values for the prediction of IR by HDL subfractions and subclasses.

Based on a comparison of the HDL subfraction profiles of the IR-free control and IR case groups, the subfractions that strongly (HDL-1 to -3) and moderately (HDL-4 to 6) reduced cardiovascular risk [31] were present in significantly lower concentrations in blood samples from the IR group. The HDL-7 to -10 subfractions considered neutral for CVD risk estimation showed no significant difference between the two groups.

The HDL-1 to -6 subfractions and HDL-L and -I subclasses showed a significant negative association with the progression and risk of IR after adjustment for relevant covariates (age, sex, ethnicity, BMI, medication, current smoking status, systolic BP, LDL and HbA1c levels). These results are in harmony with previous publications [10,29,32]. In most of these studies, only two subfractions (HDL_2_ and HDL_3_) were determined by ultracentrifugation and it was found that both native HDL-C levels and these two subfractions showed significant negative correlations with the risk of IR. However, these studies—in addition to the fact that only two subfractions were examined—had some other weaknesses, as the results were based on correlational analyses and the statistical analyses were not adjusted for relevant factors that could significantly influence carbohydrate metabolism (such as sex, age and BMI).

Of the three IR-related laboratory parameters (fasting insulin, fasting glucose and HbA1c levels) examined in the present study, native HDL-C showed a significant association with fasting insulin levels. Seven of the ten subfractions (HDL-1 to -7) showed significant associations with fasting insulin levels, while HDL-7 and -8 showed significant associations with fasting blood glucose levels. HDL-7 was the only subfraction that showed a significant association with all three laboratory parameters tested. The effects of HDL-C and its subfractions on insulin levels (and hence blood glucose) have been known for decades [33,34], and the underlying mechanisms are partially understood.

Ochoa-Guzmán and colleagues investigated the effects of HDL and its subfractions in vitro in a MIN-6 β-cell line and found that HDL particles isolated from healthy individuals promoted insulin secretion [30]. However, although no significant differences were found, a trend was observed that, at low glucose concentrations, the small and dense HDL subfractions had the strongest effect on insulin secretion, with HDL_3a_ showing the strongest correlation of all. They also observed that glucose promoted insulin secretion in a concentration-dependent manner, but that insulin secretion was further enhanced in the presence of HDL, independent of glucose levels. Studies in mice suggest that cholesterol accumulation in islet β-cells is the cause of their pathology. HDL protects beta cells from the toxic effects of glucose and IL-1β and increases insulin production. In skeletal muscle, insulin activity and glucose uptake have been shown to increase with increasing HDL levels [35].

Although the HDL-S subclass (and subfractions) did not show a significant association with either HOMA-IR levels or the development of insulin resistance in our present study, it has been shown to have beneficial effects on endothelial function, namely the ability of blood vessels to dilate or constrict in response to various stimuli. Endothelial function is largely mediated by nitric oxide (NO), a vasodilator molecule produced by endothelial nitric oxide synthase (eNOS), an enzyme found in specialized membrane domains called caveolae. It can stimulate eNOS activity and NO production by binding to specific receptors on the endothelial surface, such as the type SR-BI and ABCA1, and by providing substrates and cofactors for eNOS, such as L-arginine and tetrahydrobiopterin (BH4) [9,36,37,38]. HDL_3_ can also be oxidatively modified by 15-lipoxygenase, an enzyme that catalyzes the incorporation of oxygen into polyunsaturated fatty acids. This modification alters the lipid and protein composition of HDL_3_ and impairs its ability to activate eNOS and enhance NO production. Indeed, 15-lipoxygenase-modified HDL_3_ has been shown to decrease eNOS expression and activity, reduce intracellular cGMP levels and increase oxidative stress in endothelial cells. These effects may contribute to endothelial dysfunction and cardiovascular disease [39]. 

The present study is the first to define an optimal cut-off for HDL-C and its ten subfractions and three subclasses indicating an increased risk of IR. The HDL-L class overall, and, in this class, HDL-2, showed the strongest correlation with the existence of IR. Previous research has identified high HDL concentration as being inversely associated with cardiovascular risk [40,41,42], obesity [43] insulin resistance [44], abnormal glucose tolerance [45] and the development of T2DM [46]. There are currently no known studies that describe the association of the HDL-2 subfraction measured by Lipoprint HDL^®^ (which is not the same as the HDL_2_ and HDL_3_ subfractions identified by different methods) with the development of IR.

This study had its strengths and limitations. A major limitation of the present study was the small sample size, which may have resulted in limited statistical power. Although our results showing a correlation between IR and HDL subfraction profiles were statistically significant even after Bonferroni correction, further analyses on a larger sample of different ethnicities would be useful to confirm our conclusions.

On the other hand, the present study had several strengths. First, compared to previous studies, our study was based on more complex statistical analyses (adjusted for multiple confounders). Furthermore, the Lipoprint HDL^®^ platform used to measure HDL subfractions provides the opportunity for the high-resolution and more accurate identification of the lipoproteins and their groups associated with the development of IR.

In conclusion, the progression and risk of IR are negatively associated with changes in the HDL subfraction profile from a cardiovascular point of view. There is a significant decrease in the mean concentration of the HDL-1-6 subfractions, while the amount of HDL-7-10 subfractions remains unchanged; it results in an unfavorable lipid profile that indicates increased cardiovascular risk. We successfully identified HDL-2 and HDL-L as the two most dominant subfractions associated with IR. For both, an optimal cut-off point was determined, which may help to predict the increased risk of developing IR.

## 4. Materials and Methods

### 4.1. Study Design and Populations

A full, detailed description of the study design and data collection was provided in our previous paper [47]. Briefly, in order to understand the background of the poor health status of the Roma population compared to the Hungarian general one, especially the high prevalence of noncommunicable diseases, a health survey was designed and carried out to create a complex database for comparative and association statistical analyses. The survey consisted of three main components: a questionnaire survey, physical examinations and laboratory tests in the adult (20–64 years old) Hungarian general (HG) and Roma populations. A total of 832 participants, including 417 HG (232 women and 185 men) and 415 Roma (307 women and 108 men), were recruited during the study (in 2018/2019). Fasting blood samples were collected from participants to perform routine laboratory tests (including fasting glucose, TAG, HDL, LDL and total cholesterol measurements) and anthropometric (e.g., body height and weight), demographic (e.g., sex, ethnicity, and age), socioeconomic and health (including medication use and blood pressure measurements) data were collected.

In the present study, participants with missing anthropometric and/or laboratory parameters (20 HG and 47 Roma) and participants on lipid-lowering therapy (27 HG and 43 Roma) were excluded from further analysis.

The remaining 695 subjects (370 HG and 325 Roma) were divided into two subgroups based on their lipid profiles: (1) participants with normal lipid profiles and (2) those with reduced HDL-C. The normal lipid profile group included subjects with normal HDL-C (≥1.03 mmol/L in men and ≥1.29 mmol/L in women TAG, TC and LDL-C (126 HG and 87 Roma).

For the sample population of the present study, 277 people (115 HG and 162 Roma) with reduced HDL-C levels and 100 people with normal lipid profiles (25 HG men, 25 Roma men, 25 HG women and 25 Roma women) were selected. A further 20 people were excluded during the optimization of the datasets.

### 4.2. Analysis of HDL Subfractions

HDL is a highly heterogeneous class of lipoproteins, identified solely by the hydration density of its particles. There are several methods for the separation of HDL into subfractions. The majority of published prospective and clinical studies have used one of the proprietary laboratory tests or in-house systems available to clinicians to evaluate the use of subfractions for outcome prediction: Lipoprint HDL^®^ (gel electrophoresis), Cardio IQ^®^ (ion mobility), NMR LipoProfile^®^ (nuclear magnetic resonance) and, until recently, Vertical AutoProfile (VAP)^®^ (ultracentrifugation) [48].

For the present study, the Lipoprint HDL subfractional assay (Quantimetrix Corp., Redondo Beach, CA, USA) was used to measure HDL subfractional concentrations by polyacrylamide gel electrophoresis according to the manufacturer’s instructions. This commercial test can separate and quantify up to 10 HDL subfractions in serum or plasma based on the linear polyacrylamide gel electrophoresis method. First, 25 μL of serum was added to 3% polyacrylamide gel tubes together with 300 μL of Lipoprint HDL loading gel solution. The tubes were photo polymerized for 30 min at room temperature using Sudan Black as a lipophilic dye. Electrophoresis was performed at a constant current of 3 mA/tube for 50 min using tubes containing serum samples and the manufacturer’s quality control sample. Subfraction bands were identified by their mobility (Rf) and scanned with an ArtixScan M1 digital scanner (Microtek International Inc., Redondo Beach, CA, USA) using very-LDL (VLDL) + LDL as the start (Rf 0.0) and albumin as the end (Rf 1.0) reference point.

Ten subfractions of HDL were distinguished between the peaks of VLDL + LDL and albumin. They were grouped into three main classes: HDL-L (from HDL-1 to -3), HDL-I (from HDL-4 to -7) and HDL-S (from HDL-8 to -10) HDL subfractions. The Lipoware software LW03-v.16-134 (Quantimetrix Corp., Redondo Beach, CA, USA) was used to calculate the cholesterol concentrations of the HDL particle subsets. The relative area under the curve of the subfraction bands was multiplied by the cholesterol concentration of the samples.

### 4.3. Data Used to Identify Insulin Resistance

A limitation in estimating IR prevalence at a population level is the variety of methods/indices used to determine it. The hyperinsulinemic-euglycemic clamp technique is considered the gold standard for identifying IR [49], but this invasive method is extremely difficult to implement at the population level. A number of feasible and non-invasive methods have been identified and are available as surrogate markers, of which the HOMA-IR index is the most widely used and most suitable for quantifying IR at a population level [50,51,52]. HOMA-IR was defined as (fasting glucose (mmol/L) × fasting insulin level (mIU/L)/22.5) and individuals with a HOMA-IR value greater than 3.63 were considered to have IR and an increased risk of developing diabetes mellitus [51]. Insulin and glucose levels were determined from blood samples taken after overnight fasting. Blood glucose levels were measured by a standard enzymatic glucose oxidation method using automated laboratory equipment, while insulin levels were determined by immunoassay-based analysis.

### 4.4. Statistical Analyses

Prevalence data were compared by the χ^2^ test. Comparisons between (IR and control) subgroups were performed by Student’s unpaired *t*-test in the case of normally distributed variables and by the Mann–Whitney U-test in the case of variables with a non-normal distribution. Correlations between continuous variables were assessed by adjusted linear regression analyses, while adjusted logistic regression analyses were used in the case of binary outcome variables.

All types of regression analyses were carried out under the adjusted model (ethnicity, age, sex, BMI, current smoking status, LDL levels, HbA1c, systolic blood pressure, antihypertensive and antidiabetic treatment).

To estimate the optimal cut-off points of the HDL subfractions and subclasses for discriminating IR, the receiver operating characteristic (ROC) curve was applied. Youden’s method [53] was applied to find an optimal cut-off point on the ROC curves to optimize the sensitivity and specificity of each of them. The index was calculated for all points of the ROC curves, and the maximum value of the index was used as a criterion for selecting the optimum cut-off point. To characterize the predictive power of different biochemical parameters and surrogate indices for IR, area under the curve (AUC) calculations were carried out. Analyses were performed using the SPSS software version 26.0 (IBM, Armonk, NY, USA).

Bonferroni correction was applied for multiple analyses of the same dependent variable to avoid type I error, and the *p*-value determined by Bonferroni correction was considered as the threshold for statistical significance.

## Figures and Tables

**Figure 1 ijms-24-13563-f001:**
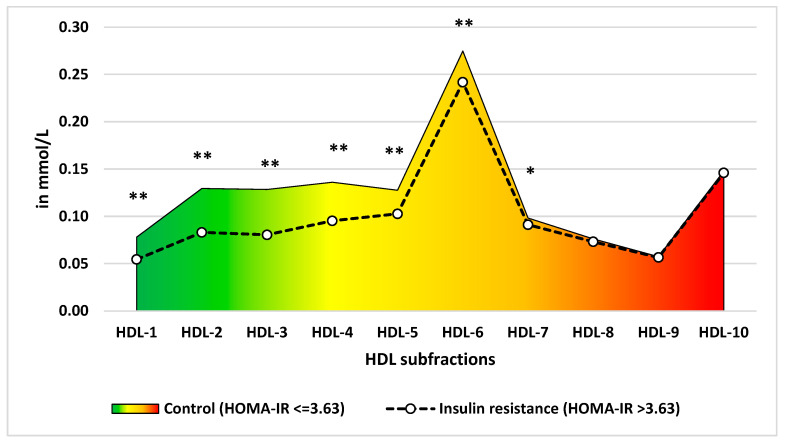
Comparison of mean high-density lipoprotein subfraction profiles between control and insulin resistance groups. HOMA-IR: Homeostasis Model Assessment of Insulin Resistance. *: *p* < 0.05, **: *p* < 0.001.

**Table 1 ijms-24-13563-t001:** Anthropometric, laboratory and demographic characteristics of the study groups.

	Control (HOMA-IR ≤ 3.63) *n* = 240	Insulin Resistance (HOMA-IR > 3.63) *n* = 137	*p*-Value
	Mean (95%CI)		
Age (in years)	40.73 (39.16–42.29)	41.47 (39.42–43.53)	0.552
BMI (kg/m^2^)	25.86 (25.15–26.56)	30.62 (29.59–31.66)	<0.001 *
Fasting glucose (mg/dL)	82.54 (80.77–84.31)	112.66 (105.90–119.41)	<0.001 *
Fasting insulin (mU/L)	8.43 (9.91–8.95)	37.92 (33.34–42.50)	<0.001 *
HbA1c (%)	5.33 (5.27–5.40)	5.77 (5.61–5.93)	<0.001 *
Fasting LDL-C (mmol/L)	2.87 (2.77–2.96)	3.16 (2.97–3.34)	0.007 *
Fasting TAG (mmol/L)	1.35 (1.24–1.47)	2.24 (2.04–2.44)	<0.001 *
Systolic BP (mmHg)	121.80 (119.77–123.84)	126.68 (124.17–129.19)	0.002 *
	Prevalence in % (95%CI)		*p*-Value
Roma ethnicity	57.08 (50.77–63.23)	54.74 (46.38–62.91)	0.660
Women	67.08 (60.96–72.80)	66.42 (58.24–73.92)	0.896
Antihypertensive treatment	21.25 (16.44–26.75)	34.31 (26.75–42.52)	0.005 *
Antidiabetic treatment	2.92 (1.31–5.64)	14.60 (9.45–21.23)	<0.001 *
Current smoker	56.49 (50.15–62.66)	50.74 (42.39–59.05)	0.282

HOMA-IR: Homeostasis Model Assessment of Insulin Resistance, SD: standard deviation, 95%CI: 95% confidence interval, BMI: body mass index, HbA1c: hemoglobin A1c, LDL: low-density lipoprotein, TAG: triacylglycerol, BP: blood pressure, *: *p* < 0.05.

**Table 2 ijms-24-13563-t002:** The association of native high-density lipoprotein cholesterol (HDL-C) levels and its subfractions (HDL-1 to -10) and subclasses (HDL-L, -I and -S) with fasting insulin, fasting glucose and hemoglobin A1c (HbA1c) levels.

	Fasting Insulin Level (U/dL)		Fasting Glucose Level (mmol/L)		HbA1c (%)	
	B (SE)	*p*-Value	B (SE)	*p*-Value	B (SE)	*p*-Value
HDL-C	−16.55 (2.79)	7.45 × 10^−9^ **	0.28 (0.23)	0.224	−0.22 (0.10)	0.028 *
HDL-1	−120.25 (27.39)	1.50 × 10^−5^ **	3.66 (2.14)	0.088	−2.52 (0.95)	0.008 *
HDL-2	−104.92 (16.42)	5.65 × 10^−10^ **	0.51 (1.37)	0.711	−1.05 (0.61)	0.087
HDL-3	−78.47 (15.41)	5.81 × 10^−7^ **	−0.60 (1.24)	0.629	−0.67 (0.55)	0.229
HDL-4	−72.03 (16.76)	2.20 × 10^−5^ **	−2.12 (1.31)	0.106	0.25 (0.60)	0.669
HDL-5	−123.69 (26.09)	0.30 × 10^−6^ **	−2.07 (2.07)	0.318	−0.65 (0.93)	0.485
HDL-6	−72.75 (15.13)	2.00 × 10^−6^ **	2.65 (1.19)	0.026 *	−1.45 (0.53)	0.006 *
HDL-7	−141.19 (36.48)	1.29 × 10^−4^ **	10.30 (2.74)	1.96 × 10^−4^ **	−3.74 (1.24)	0.003 **
HDL-8	−129.89 (45.31)	0.004 **	11.92 (3.35)	4.23 × 10^−4^ **	−3.28 (1.54)	0.033 *
HDL-9	−71.28 (56.85)	0.211	8.37 (4.23)	0.049 *	−1.50 (1.91)	0.434
HDL-10	−14.80 (19.25)	0.443	1.73 (1.43)	0.229	−0.54 (0.64)	0.406
Large HDL	−38.72 (6.64)	1.23 × 10^−8^ **	0.21 (0.54)	0.696	−0.46 (0.24)	0.059
Intermediate HDL	−30.06 (5.65)	1.80 × 10^−7^ **	0.44 (0.45)	0.331	−0.40 (0.20)	0.049 *
Small HDL	−16.40 (12.21)	0.180	1.93 (0.91)	0.034 *	−0.55 (0.41)	0.182

B: unstandardized coefficient, SE: standard error, *: *p* < 0.05, **: *p* < 0.005 (Bonferroni-corrected *p*-value).

**Table 3 ijms-24-13563-t003:** The association of native high-density lipoprotein cholesterol (HDL-C) levels and its subfractions (HDL-1 to -10) and subclasses (HDL-L, -I and -S) with Homeostasis Model Assessment of Insulin Resistance (HOMA-IR) and presence of insulin resistance (HOMA-IR ≥ 3.63).

	HOMA-IR		Insulin Resistance (HOMA-IR ≥ 3.63)	
	B (SE)	*p*-Value	B (SE)	*p*-Value
HDL-C	−5.75 (1.01)	2.41 × 10^−8^ **	−1.91 (0.45)	2.00 × 10^−5^ **
HDL-1	−37.00 (9.99)	2.45 × 10^−4^ **	−8.66 (3.98)	0.030 *
HDL-2	−40.37 (5.83)	2.08 × 10^−11^ **	−11.26 (2.65)	2.20 × 10^−5^ **
HDL-3	−32.95 (5.41)	3.01 × 10^−9^ **	−8.53 (2.34)	2.67 × 10^−4^ **
HDL-4	−35.07 (5.82)	4.14 × 10^−9^ **	−8.94 (2.51)	3.70 × 10^−4^ **
HDL-5	−54.37 (9.10)	5.73 × 10^−9^ **	−14.85 (3.93)	1.16 × 10^−4^ **
HDL-6	−20.70 (5.53)	2.14 × 10^−4^ **	−7.83 (1.83)	8.47 × 10^−4^ **
HDL-7	−21.73 (13.33)	0.104	−11.50 (5.41)	0.034 *
HDL-8	−10.98 (16.43)	0.504	−11.22 (6.64)	0.091
HDL-9	−0.95 (20.55)	0.963	−9.27 (8.29)	0.264
HDL-10	−0.01 (6.90)	0.999	−2.51 (2.77)	0.364
Large HDL	−14.85 (2.36)	9.52 × 10^−10^ **	−3.85 (1.03)	1.99 × 10^−4^ **
Intermediate HDL	−10.66 (2.03)	2.80 × 10^−7^ **	−3.43 (0.89)	1.13 × 10^−4^ **
Small HDL	0.09 (4.40)	0.984	−2.37 (1.78)	0.183

B: unstandardized coefficient, SE: standard error, *: *p* < 0.05, **: *p* < 0.005 (Bonferroni-corrected *p*-value).

**Table 4 ijms-24-13563-t004:** The optimal cut-off points for native high-density lipoprotein cholesterol (HDL-C) and its subfractions (HDL-1 to 10) and subclasses (HDL-L, -I and -S) based on Youden’s J statistic.

	AUC	Sens./Spec.	Youden Index	Optimal Cut-Off Point (in mmol/L)
HDL-C	0.715	0.696/0.628	0.324	1.045
HDL-1	0.676	0.646/0.679	0.325	0.057
HDL-2	0.722	0.579/0.796	0.375	0.102
HDL-3	0.715	0.621/0.737	0.358	0.093
HDL-4	0.709	0.567/0.803	0.370	0.113
HDL-5	0.711	0.692/0.672	0.363	0.106
HDL-6	0.661	0.633/0.620	0.254	0.250
HDL-7	0.584	0.796/0.365	0.161	0.082
HDL-8	0.540	0.821/0.285	0.106	0.060
HDL-9	0.504	0.850/0.234	0.084	0.043
HDL-10	0.507	0.296/0.752	0.048	0.173
Large HDL	0.723	0.604/0.803	0.407	0.264
Intermediate HDL	0.699	0.546/0.781	0.327	0.584
Small HDL	0.518	0.625/0.431	0.056	0.253

AUC: area under the curve, Sens.: sensitivity, Spec.: specificity.

**Table 5 ijms-24-13563-t005:** Correlation of native high-density lipoprotein cholesterol (HDL-C) level below the optimal and its subfractions (HDL-1 to -10) and subclasses (HDL-L, -I and -S) with the existence of insulin resistance (Homeostasis Model Assessment of Insulin Resistance ≥ 3.63).

	HOMA-IR		Insulin Resistance (HOMA-IR ≥ 3.63)	
	B (SE)	*p*-Value	OR (95%CI)	*p*-Value
HDL-C (<1.045 mmol/L)	3.22 (0.70)	5.00 × 10^−6^ **	2.94 (1.71–5.06)	1.01 × 10^−4^ **
HDL-1 (<0.057 mmol/L)	2.48 (0.73)	7.05 × 10^−4^ **	2.08 (1.21–3.59)	0.009 *
HDL-2 (<0.102 mmol/L)	4.39 (0.73)	4.02 × 10^−9^ **	3.81 (2.06–7.02)	1.80 × 10^−5^ **
HDL-3 (<0.093 mmol/L)	3.49 (0.73)	3.00 × 10^−6^ **	2.87 (1.62–5.10)	3.09 × 10^−4^ **
HDL-4 (<0.113 mmol/L)	3.68 (0.69)	1.62 × 10^−7^ **	3.80 (2.12–6.80)	7.00 × 10^−6^ **
HDL-5 (<0.106 mmol/L)	3.24 (0.68)	3.00 × 10^−6^ **	3.03 (1.78–5.16)	4.3 × 10^−5^ **
HDL-6 (<0.250 mmol/L)	2.37 (0.67)	4.33 × 10^−4^ **	2.37 (1.40–4.02)	0.001 **
HDL-7 (<0.082 mmol/L)	1.51 (0.73)	0.039 *	2.06 (1.19–3.56)	0.010 *
HDL-8 (<0.060 mmol/L)	1.85 (0.80)	0.021 *	2.22 (1.20–4.10)	0.011 *
HDL-9 (<0.043 mmol/L)	1.10 (0.85)	0.198	2.00 (1.05–3.83)	0.036 *
HDL-10 (<0.173 mmol/L)	−0.09 (0.73)	0.905	1.28 (0.73–2.24)	0.393
Large HDL (<0.264 mmol/L)	4.57 (0.73)	1.09 × 10^−9^ **	4.73 (2.53–8.86)	1.00 × 10^−6^ **
Intermediate HDL (<0.584 mmol/L)	3.50 (0.65)	1.55 × 10^−7^ **	3.47 (1.98–6.06)	1.30 × 10^−5^ **
Small HDL (<0.253 mmol/L)	0.08 (0.69)	0.905	1.63 (0.95–2.77)	0.075

B: unstandardized coefficient, SE: standard error, *: *p* < 0.05, **: *p* < 0.005 (Bonferroni-corrected *p*-value).

## Data Availability

Data are available on request due to privacy or ethical concerns.

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
