# Peer review of "Association of HDL Subfraction Profile with the Progression of Insulin Resistance"

_ijms, 2023, doi:10.3390/ijms241713563_

Round 1

Reviewer 1 Report

In this manuscript, Piko et al describe the association between HDL subfractions and insulin resistance. They classified HDL into 10 fractions and characterized the correlation of each fraction with HOMA-IR. While the association between large HDL subfractions and insulin resistance is not completely novel, their detailed analysis of 10 subfractions has an important implication to understand the significance of HDL subfractions. To strengthen their argument, I would raise the following points:

1.  Can you provide a more detailed definition of each HDL subfraction? Particularly, describing which subfractions in your assay correspond to conventionally measured HDL-2/-3 fractions is necessary to avoid confusion.

2. If you divide the entire study population into obese and non-obese groups, do you observe similar differences in HDLC subfractions as in Fig. 1?

3. Suppl Table 1 contains very intriguing information since different HDL subfractions are correlated with insulin levels, glucose, and HbA1c levels, respectively. I would recommend including this in a main table and providing more discussion regarding how different HDLC subfractions are correlated with glucose and A1c (lines 188-201).

4. In the material section, measurement methods for glucose and insulin, as well as calculation for HOMA-IR should be included. 

5. Did you exclude subjects with diabetes?

Reviewer 2 Report

The present study defined an optimal cut-off for HDL-C and its ten subfractions and three subclasses indicating an increased risk of IR. HDL-L class overall and the HDL-2 subfraction showed the strongest correlation with the existence of IR. This study is very interesting, but needs additional data and discussion before it can be accepted.

Major comments

1. Insulin resistance is closely related to triglycerides. Please add data for triglycerides.

2. Please add a discussion on the generation and metabolic processes of HDL subfractions. For example, please discuss the relationship between HDL-1-3 and ABCA1, CETP, LPL, HTGL and mechanisms of cholesterol abstraction. Also, please discuss the relationship between HDL-4-7 and HDL-8-10 and ABCA1, CETP, LPL, HTGL and mechanisms of cholesterol withdrawal.

3. The authors cite past reports as follows in the Introduction.

“Generally, larger, and less dense HDL particles are more anti-inflammatory and anti-atherogenic than smaller and denser ones. For example, large HDL (HDL-L) has a higher capacity to remove cholesterol from macrophages and inhibit LDL oxidation than intermediate-HDL (HDL-I) or small-HDL (HDL-S) [11]. However, some studies have also suggested that smaller HDL particles may have beneficial effects on endothelial function and nitric oxide production [12]. The profile of HDL subfractions may vary depending on environmental-, lifestyle- [13], genetic- factors [14] and metabolic disorders [15].”

Please add a discussion about the physiological role of HDL-S based on the results of this study.

4. The authors cite past reports as follows in the Discussion.

“Ochoa-Guzmán and colleagues investigated the effects of HDL and its subfractions in vitro in a MIN-6 β-cell line and found that HDL particles isolated from healthy individuals promote insulin secretion [21].”

Please discuss which subfractions of HDL promote β-cell insulin secretion.

Minor comment.

Please clearly state what the vertical axis in Figure 1 indicates.

Round 2

Reviewer 1 Report

The authors sufficiently addressed my questions. The revised manuscript is suitable for publication.